# Effects of Coronavirus Persistence on the Genome Structure and Subsequent Gene Expression, Pathogenicity and Adaptation Capability

**DOI:** 10.3390/cells9102322

**Published:** 2020-10-19

**Authors:** Ching-Hung Lin, Cheng-Yao Yang, Meilin Wang, Shan-Chia Ou, Chen-Yu Lo, Tsung-Lin Tsai, Hung-Yi Wu

**Affiliations:** 1Graduate Institute of Veterinary Pathobiology, College of Veterinary Medicine, National Chung Hsing University, Taichung 40227, Taiwan; tw23whale@hotmail.com (C.-H.L.); yangchengyao@nchu.edu.tw (C.-Y.Y.); axfbji7917@gmail.com (C.-Y.L.); windtaker10@msn.com (T.-L.T.); 2Department of Microbiology and Immunology, School of Medicine, Chung-Shan Medical University, Taichung 40201, Taiwan; wml@csmu.edu.tw; 3Graduate Institute of Microbiology and Public Health, College of Veterinary Medicine, National Chung Hsing University, Taichung 40227, Taiwan; scou@dragon.nchu.edu.tw

**Keywords:** coronavirus, persistence, genome structure, gene expression, pathogenicity, virus fitness

## Abstract

Coronaviruses are able to establish persistence. However, how coronaviruses react to persistence and whether the selected viruses have altered their characteristics remain unclear. In this study, we found that the persistent infection of bovine coronavirus (BCoV), which is in the same genus as SARS-COV-2, led to alterations of genome structure, attenuation of gene expression, and the synthesis of subgenomic mRNA (sgmRNA) with a previously unidentified pattern. Subsequent analyses revealed that the altered genome structures were associated with the attenuation of gene expression. In addition, the genome structure at the 5′ terminus and the cellular environment during the persistence were responsible for the sgmRNA synthesis, solving the previously unanswered question regarding the selection of transcription regulatory sequence for synthesis of BCoV sgmRNA 12.7. Although the BCoV variants (BCoV-p95) selected under the persistence replicated efficiently in cells without persistent infection, its pathogenicity was still lower than that of wild-type (wt) BCoV. Furthermore, in comparison with wt BCoV, the variant BCoV-p95 was not able to efficiently adapt to the challenges of alternative environments, suggesting wt BCoV is genetically robust. We anticipate that the findings derived from this fundamental research can contribute to the disease control and treatments against coronavirus infection including SARS-CoV-2.

## 1. Introduction

Coronaviruses (CoVs), which are single-stranded, positive-sense RNA viruses with a genome size of 26–32 kilobases (kb), belong to the subfamily *Coronavirinae*, family *Coronaviridae*, order *Nidovirales* [1,2,3]. The structure of the coronavirus genome consists of a 5′ cap, a 5′-untranslated region (UTR), open reading frames (ORFs), and a 3′-UTR including a poly(A) tail. The high-ordered structures in the 5′ and 3′ UTR have been identified as *cis*-acting RNA elements associated with coronavirus gene expression. The 5′ two-thirds of the genome encode replicase-related nonstructural proteins (nsps), and the other one-third of the genome mostly encodes structural proteins [2,4]. In addition to the replication of genomic RNA in infected cells, a nested set of subgenomic mRNAs (sgmRNAs), which are 5′ and 3′ coterminal with the genome [2,4], are synthesized using the genome as a template at the transcription regulatory sequence (TRS). However, due to the lack of packaging signal in sgmRNAs, the budding virus particles only contain genome [5]. The synthesis of sgmRNAs is a feature of Nidoviruses and defined as transcription. TRS is a sequence motif located at the downstream of the leader sequence (TRS-L) and upstream of each structural or accessory gene in the genome body (TRS-B). The TRS contains a ~6 nt conserved sequence (CS) flanked by variable sequences at its 5′ (5′ TRS) and 3′ (3′ TRS) ends [6]. Because TRS-B shows complementarity to cTRS-L, sgmRNA can be synthesized through base pairing by similarity-assisted homologous recombination mechanism [6,7,8,9]. Many factors including complementarity between the cTRS-L and the TRS-B, TRS secondary structure, and viral and cellular proteins have been shown to affect the levels of sgmRNA synthesis. Bovine coronavirus (BCoV), which is in the genus *betacoronavirus* [10], contains three TRSs (designated TRS1, TRS2, and TRS3) located upstream of 12.7 kDa gene [11]. Because TRS1, TRS2, and TRS3 show different levels of complementarity to the 5′ leader TRS (cTRS-L), they can all be potentially used to synthesize sgmRNA 12.7 kDa (sgmRNA 12.7) through base pairing by a similarity-assisted homologous recombination mechanism in infected cells. However, although complementarity is higher between the cTRS-L and the TRS1-B, TRS2-B is mostly employed to synthesize sgmRNA 12.7 kDa (sgmRNA 12.7) in infected cells. It has been suggested that the sequence downstream of the TRS2 can affect the choice of TRS for the sgmRNA 12.7 synthesis [12]. However, the mechanism leading to this outcome remains unknown.

In addition to acute coronavirus infection, several lines of evidence have shown that coronaviruses including SARS-CoV-2 are able to establish persistence in vitro and in vivo [13,14,15,16,17,18]. Viral persistence is a coevolution process between the virus and host cells during which the virus adapts the environment via alterations of genome structures and host cells resist the viral cytopathology [19,20]. Understanding the mechanism by which coronavirus establishes persistence and subsequent virus fitness may reveal viral gene functions and their effects on pathogenicity, thus demonstrating medical importance [21,22]. During persistent BCoV infection, intraleader RNA mutation in subgenomes has been observed and is associated with translation attenuation of the downstream ORF [23]. However, variations at the 5′ terminal sequence in mouse hepatitis virus-A59 (MHV-A59), a mouse coronavirus, have not been identified, but instead a mutation at nucleotide (nt) 77 in the genome leads to the synthesis of the start codon AUG and subsequent translation enhancement [18]. The recently identified leaderless genome in BCoV has also been suggested to be associated with an attenuation of translation during BCoV persistence [24]. These studies focus only on the structural changes at the 5′ end of the genome and subgenome during persistence [18,23,24]. Alterations of the whole genome structure in coronaviruses during persistence, however, have not been documented. In addition, whether the alterations in genome structure during persistence affect coronavirus pathogenicity in different host cells remains unclear. With the unique genetic organization and replication strategy of coronavirus [2], understanding the molecular mechanisms leading to persistence and characterizing the features of the resultant coronavirus may unveil unknown viral features of medical importance.

Coronaviruses including SARS-CoV-2 are able to establish persistence [13,14,15,16,17,18]. However, how coronaviruses response to persistence and whether the selected viruses through the persistence have altered their characteristics such as pathogenicity are important questions that remain unanswered. In this study, BCoV, which is in the same genus (*betacoronavirus*) as SARS-COV-1, SARS-CoV-2, and MERS-CoV, was used to explore the effects of persistence on the genome structure and subsequent gene expression, pathogenicity, and adaptation capability. The results derived from the study may enable us to further understand the mechanism of gene expression and coronavirus infection, thus contributing to the disease control and treatments for coronaviruses.

## 2. Materials and Methods

### 2.1. Viruses and Cells

BCoV strain of Mebus (GenBank accession no. U00735), which was obtained from David A. Brian (University of Tennessee, TN), was plaque-purified [25,26] and grown in human rectum tumor (HRT)-18 cells [27]. Mouse L (ML) cells and HRT-18 cell were also obtained from David A. Brian and maintained in Dulbecco’s modified Eagle’s medium (DMEM) supplemented with 10% fetal bovine serum (FBS) (Hyclone, UT, USA) at 37 °C with 5% CO_2_.

### 2.2. Establishment of Persistence and Adaptation

For the establishment of coronavirus persistence, HRT-18 cells were infected with wild type (wt) BCoV at a multiplicity of infection (MOI) of 1. The surviving cells (<5%) at 3 days postinfection (dpi) were thereafter passaged at every fourth day. The virus and cellular RNA were collected at each passage and the virus collected at 95 days (d) of persistent infection was designated BCoV-p95. For the experiment of BCoV-p95 in fresh HRT-18 cells, HRT-18 cells were infected with wild-type (wt) BCoV or BCoV-p95 at a MOI of 1 for VP0-VP10. For the adaptation experiment of BCoV-p95 in HRT-18 cells treated with polyinosinic:polycytidylic acid (poly IC), HRT-18 cells were first treated with poly IC with final concentration of 1 μg/mL and after 4 h of treatment, HRT-18 cells were infected with wt BCoV or BCoV-p95 at a MOI of 1. The virus was collected at 48 hpi and virus at this stage is defined as virus passage 0 (VP0). The titer was 10^7.7^ pfu/mL for wt BCoV; however, almost no virus plaque could be detected for BCoV-p95 (10^1.4^ pfu/mL) at VP0. In order to determine whether BCoV-p95 could gradually adapt to the challenge, we performed blind passage until VP10. On the other hand, the collected wt BCoV at VP0 was passaged to fresh HRT-18 cells (VP1) at a MOI of 1 in the presence of poly IC with final concentration of 1 μg/mL. The wt BCoV passage step was repeated until VP10. At each passage of virus, virus, cellular RNA and lysates were collected for the subsequent assays. In addition, for the adaptation experiments of BCoV-p95 and wt BCoV in ML cells, ML cells were respectively infected with the aforementioned viruses at a MOI of 1 for the first time of infection. The titer was 10^5.2^ pfu/mL for wt BCoV and 10^3.8^ pfu/mL for BCoV-p95 at VP0. Therefore, for VP1-VP10 infection, 0.001 MOI was used for both viruses. At VP10, virus, cellular RNA, and lysates were collected for subsequent assays.

### 2.3. Identification of Genome Structure

To determine the terminal sequence including poly(A) tail length for BCoV genomic RNA, a head-to-tail ligation method was employed as described previously [28]. The head-to-tail ligated RNA was used for reverse transcription (RT) with SuperScript III reverse transcriptase (Invitrogen, Carlsbad, CA, USA) and PCR was performed with AccuPrime Taq DNA polymerase (Invitrogen, Carlsbad, CA, USA) and oligonucleotides binding to 5′ and 3′ UTR of the genome followed by sequencing. The structures of the *cis*-acting RNA located in the 5′ UTR were predicted using the Mfold algorithm [29]. To determine whether the genome structures of BCoV were altered under the persistence, random hexamer oligonucleotides were used for RT with SuperScript III reverse transcriptase (Invitrogen, Carlsbad, CA, USA), and the resulting cDNA was used for PCR with PfuUltra II high-fidelity DNA polymerase (Agilent, Santa Clara, CA, USA). The resultant PCR products were then subject to sequencing analysis. Because conventional Sanger sequencing was used, the sequences shown in the passaging experiments represented the genome structure of the main virus population but not the quasispecies. For the identification of the TRS employed for the synthesis of sgmRNAs, oligonucleotides binding to the leader sequence, and the sequence downstream of the TRS of each sgmRNA were used followed by sequencing.

### 2.4. Construction of Plasmid and In Vitro Transcription

An overlap PCR was employed to create defective interfering (DI) RNA mutants derived from wt DI RNA and has been described previously [30]. In brief, constructs pDI-73C, pDI-TAA, pDI-D10, pDI-D5G and pDI-D4U for replication assay were generated using pDI-Wt DNA as the template with appropriate oligonucleotides containing various mutated 5′-terminal sequences corresponding to the constructs for PCR. The resulting PCR product was digested with *NgoMIV* and *XbaI* and cloned into *NgoMIV*- and *XbaI*- linearized pDI-Wt to create the aforementioned constructs. Constructs for translation assay were similarly generated but using constructs pDI-Wt with EGFP gene as described previously [31]. The construction of pSgm-DI-12.7 for the transcription assay has been described previously [12] and thus to construct pSgm-DI-73C-12.7, the 5′ end of pDI-73C was digested with *NgoMIV* and *XbaI* and cloned into *NgoMIV*- and *XbaI*- linearized pSgm-DI-12.7. The overlap PCR was also applied to generate pS-DI-E and S-DI73C-E and the PCR product was digested with *NgoMIV* and *MluI* and cloned into *NgoMIV*- and *MluI*- linearized pDI-Wt. For in vitro transcription, capped transcripts using *MluI*-linearized plasmid DNA constructs as templates were prepared with a T7 mMessage mMachine kit (Thermo Fisher Scientific, Waltham, MA, USA) according to the manufacturer’s protocol.

### 2.5. Determination of Virus Titer

The plaque assay for BCoV was performed in 6-well Costar plates (Costar, Cambridge, MA, USA.) [32]. Virus with serial dilution was added into HRT-18 cells and at 1 hpi, HRT-18 cells were washed with DMEM followed by an agarose overlay containing DMEM, 0.6% agarose, and 2% FBS. HRT-18 cells were then incubated at 37 °C with 5% CO_2_ for 72f h and viral plaques were visualized by haemadsorption with mouse red blood cells. The virus titer was determined by scoring the number of haemadsorption foci.

### 2.6. Northern Blot Assay

Ten μg of total cellar RNA was electrophoresed through a formaldehyde-agarose gel. By vacuum blotting, RNA was transferred from the gel to a Nytran membrane, and BCoV RNA was detected with the oligonucleotide, which was 5′-end labeled with ^32^P and bound to BCoV 3′ UTR. The probed blot was exposed to Kodak XAR-5 film (Kodak, Rochester, NY, USA).

### 2.7. Western Blot Assay

The collected cell lysates were separated using 12% sodium dodecyl sulphate-polyacrylamide gel electrophoresis (SDS-PAGE) gels and electrotransferred onto nitrocellulose membranes (GE Healthcare, Chicago, IL, USA). An antibody against BCoV nsp1, N protein, enhanced green fluorescent protein (EGFP) or β-actin was used as the primary antibody followed by goat anti-mouse IgG conjugated to horseradish peroxidase (HRPO) as the secondary antibody (Jackson Laboratory, Bar Harbor, ME, USA). Detected protein(s) was visualized using a Western Lightning™ Chemiluminescence Reagent (Perkin Elmer, Waltham, MA, USA) and Kodak XAR-5 film (Kodak, Rochester, NY, USA).

### 2.8. RT-qPCR

To measure the synthesis of genome and subgenome, 2 μg of TRIzol-extracted total cellular RNA was used for the RT reaction. For measurement of genome synthesis, oligonucleotides binding to the leader sequence and sequence in the nsp1 gene were used. For measurement of subgenome synthesis, oligonucleotides binding to the leader sequence and sequence downstream of TRS for each subgenome gene were used. For comparison of the replication efficiency between DI RNAs with 5′ terminal sequence alterations, oligonucleotides binding to N gene and reporter gene of DI RNA were employed to differentiate the DI RNA from the helper virus. For comparison of the efficiency of sgubgenomic DI RNA synthesis between the constructs Sgm-DI-12.7 and Sgm-DI-73C-12.7, oligonucleotides binding to the leader sequence and reporter gene downstream of TRS were employed to differentiate the sgm DI RNA from the helper virus as described previously [12]. For qPCR, SYBR^®^ green amplification mix (Roche Applied Science, Mannheim, Germany) and oligonucleotides were used according to the manufacturer’s protocol. In these experiments, dilutions of plasmids containing the same gene as the detected genome, subgenomes, DI RNA, or sgm DI RNA were always run in parallel with the quantitated cDNA for use in standard curves (dilutions ranged from 10^8^ to 10 copies of each plasmid). The amount of synthesized RNA was normalized to the levels of internal controls, including helper virus genomic RNA, 18S rRNA, or input DI RNA where is needed.

### 2.9. Statistical Analysis

Data were analyzed with Student’s *t* test using Prism 6.0 software (GraphPad Software Inc., San Diego, CA, USA). The values in the study are presented as the means ± standard deviations (SD) (*n* = 3); * *p* < 0.05, ** *p* < 0.01 and *** *p* < 0.001.

## 3. Results

### 3.1. Alterations of Genome Structure in BCoV during Persistence

To determine whether the genome structure of BCoV was altered after the establishment of viral persistence, the BCoV RNA genome and subgenomes from HRT-18 cells after 95 days (d) of persistent infection were sequenced and analyzed. In comparison to wild type (wt) BCoV, the sequence alterations of both nucleotides (nt) and amino acids (aa) were identified and are illustrated in Figure 1 and Appendix A. Specifically, the 5′-most sequences of the genome showed heterogeneity, and such alterations also affected the 5′ cis-acting RNA structure stem-loop (SL) I and the corresponding free energy (∆G) after 95 d of infection (Figure 1B–F). A U to C substitution at nt 73, located immediately downstream of the core sequence of transcription regulatory sequence (TRS), was also identified. Such a substitution also altered the cis-acting RNA structures SL III and decreased the ∆G (Figure 1G). The sequence and cis-acting RNA structure in 3′ UTR were not altered; however, in comparison to the various poly(A) tail lengths of ~38, ~63, ~52 and ~38 nt in fresh HRT-18 cells infected with wt BCoV at 2, 8, 24, and 48 h postinfection (hpi), respectively, the poly(A) tail length became shorter (~25 nt) and showed no variations at 2, 8, 24, and 48 h after the final passage of HRT-18 cells with persistent wt BCoV infection (95 d) (Figure 1H). For the structure of subgenomic RNA (sgmRNA), the sequencing results showed that TRS used for synthesis of each sgmRNA was not changed with the exception of sgmRNA 12.7, where TRS1, instead of TRS2 (with which sgmRNA 12.7 was synthesized in HRT-18 cells freshly infected with wt BCoV), was employed during persistence (Figure 1I; sequencing data are shown in Appendix A). In addition, nt substitutions were found within the open reading frames (ORFs) of the genome (Appendix A) with a higher intensity in the nsp16 and S protein genes. Consequently, the aa substitutions were also identified (Figure 1J–L) and mostly found in nsp16 and S proteins. For nsp16 (Figure 1K), in addition to 28 aa substitutions, the presence of a nt substitution in the nsp16 gene from G to A at nt 21,287 was identified, resulting in a stop codon, and leading to a deletion of 69 aa in the C-terminal of nsp16 protein. For S protein (Figure 1L), the aa substitutions occurred mostly in the S1 subunit (20 out of 26 aa) during persistence. Together, the structures of both genome (at 5′ and 3′ termini and within the ORF, mostly in nsp16 and S protein) and subgenome 12.7 (with an alternative TRS) were altered during persistence.

### 3.2. The Gene Expression of BCoV was Attenuated during Persistence

To determine whether BCoV replication was altered during persistent infection, a plaque assay was performed to determine the virus titer. The virus collected from HRT-18 cells after 95 d of BCoV infection was designated BCoV-p95. The viral titer of BCoV-p95 was high (10^7.34^ pfu/mL, Figure 2A) but still 10-fold lower than that of wt BCoV (10^8.35^ pfu/mL, Figure 2A). In addition to the detection of virus particle number, Northern blot analysis was employed to examine the synthesis of coronaviral RNA. The sgmRNAs nucleocapcid (N), membrane (M) and envelope (E) collected from HRT-18 cells with 95 d of wt BCoV infection were detected; surprisingly, the sgmRNAs 12.7 kDa (12.7), spike (S), hemagglutinin/esterase (HE) and 32 kDa (32K) were not detectable (Figure 2B). This pattern of sgmRNA synthesis has not been previously identified in coronavirus infection. Note that the signal for BCoV genome detected by Northern blotting is generally weak. Thus, to further determine whether the genome along with the aforementioned sgmRNAs, which were not detectable by Northern blot analysis, were synthesized during persistence, RT-qPCR was performed. As shown in Figure 2C, the genome and sgmRNAs 32K, HE, S, and 12.7 collected after 95 d of persistent infection (p95) could be detected, but their amounts were ~20–50-fold lower when compared with those collected after 48 h of wt BCoV infection (Wt). Western blot analysis was also performed to determine the translation efficiency of the genome (represented by nsp1) and subgenome (represented by N protein) during persistence. As shown in Figure 2D, the translation efficiency of the nsp1 and sgmRNA N after 95 d of persistence was decreased in comparison to that after 48 h of infection. Thus, the efficiency of gene expression for BCoV was decreased during persistence.

### 3.3. Alterations in the Genome Structure are Associated with Attenuation of Gene Expression during Persistence

It has been well characterized that the 5′ and 3′ termini of the coronavirus genome harbor cis-acting RNA elements that are required for efficient gene expression [2]. To determine whether the alterations in the 5′ and 3′ termini of the genome and subgenomes (Figure 1) were connected to the attenuation of gene expression during persistence, BCoV defective interfering (DI) RNA (Figure 3A), which is a surrogate for the BCoV genome and has been extensively used to study cis-acting RNA elements required for coronavirus gene expression, was employed for this aim [7,8,9,31,34,35,36]. To determine whether the replication efficiency was affected by the alterations of 5′-terminal sequences, DI RNA was engineered to contain the 5′-modified sequences (Figure 3B) identified during persistence (Figure 1C–G). The resulting DI RNA constructs were respectively transfected into wt BCoV-infected HRT-18 cells followed by RT-qPCR. The replication efficiency between DI RNA constructs was variable (Figure 3C). That is, in comparison with wt DI RNA (DI-Wt), DI-73C and DI-TAA showed similar replication efficiency, but DI-D10, DI-D5G and DI-D4U displayed reduced replication efficiency. Regarding the translation efficiency, to avoid the effect of replication on translation efficiency, DI RNA constructs carrying the EGFP gene (Figure 3D) were respectively transfected into uninfected cells, and as shown in Figure 3E, the translation efficiency of DI RNA constructs was similar (DI-TAA, DI-D5G and DI-D4U) or higher (DI-73C and DI-D10) in comparison with that of wt DI RNA (DI-Wt). These results suggest that the 5′-terminal heterogeneous sequences synthesized during persistence may not play important roles in the decrease of translation efficiency. However, it has been demonstrated that DI RNA with shorter poly(A) tail length show decreased replication and translation efficiency [28,37]. Thus, DI RNA with a poly(A) length of 25 nt during persistence (Figure 1H) may display decreased efficiency of gene expression in comparison to that with a poly(A) length longer than 25 nt during acute infection (Figure 1H). These results together suggest that the altered 5′ and 3′ terminal sequences caused by persistence are associated with the decreased gene expression of DI RNA.

Since a mutation from U to C at nt 73 within the leader TRS altered the secondary structure of SL III, it was hypothesized that such mutation may affect the selection of the body TRS and possibly the efficiency of sgmRNA 12.7 synthesis. To test this possibility, DI RNA with or without the C mutation at nt 73 was engineered to contain sgmRNA12.7 TRS1, TRS2, and TRS3 (TRS3 is not shown in color) (Figure 3F, left panel). The results showed that TRS1 was still selected for synthesis of sgmRNA DI 12.7 (sgm-DI-73C-12.7) derived from DI-73C-12.7 (C mutation at nt 73) (Figure 3F, left panel; the sequencing data not shown). By RT-qPCR, it was found that the efficiency of sgm-DI-73C-12.7 synthesis was lower than that of sgm-DI-12.7 (Figure 3F). However, the translation efficiency was similar between synthesized sgm DI RNAs with TRS1 (S-DI-E) or TRS2 (S-DI73C-E) (Figure 3G). Therefore, the mutation at nt 73 in the leader TRS of the genome affected the selection of TRS and subsequent reduced efficiency of sgmRNA 12.7 synthesis.

Taken together, the results suggest that the alterations of genome structures in cis-acting elements located at 5′ and 3′ termini and leader TRS (Figure 1B–I) caused by BCoV persistence can decrease the efficiency of gene expression. Thus, the alterations in genome structures are associated with the attenuation of gene expression during persistence, as shown in Figure 2.

### 3.4. The BCoV-p95 Derived from Persistence Replicates Efficiently in Regular Host HRT-18 Cells

Under the selection pressure of persistence, the structures of the BCoV genome and gene expression were altered (Figure 1 and Figure 2), and these modulations might represent the means for coronavirus to adapt to the environment with persistent infection. We further addressed whether BCoV-p95 with an altered genome structure caused by persistent infection showed modified biological characteristics when returned to its regular host HRT-18 cells without selection pressure. For this, sequence analysis of BCoV-p95 virus stock collected from the supernatant of persistent infection cells was first performed to ensure that the genome structure from the supernatant viruses was not altered in comparison with that from persistently infected cells, and fresh HRT-18 cells were then respectively infected with wt BCoV and BCoV-p95. After 48 h of infection, virus was collected and virus at this stage is defined as virus passage 0 (VP0). The virus viters during VP0-VP10 for BCoV-p95 and wt BCoV were shown in Appendix A. Both wt BCoV (wt) and BCoV-p95 (p95) grew to high titers (10^8.4^ pfu/mL and 10^7.7^ pfu/mL, respectively) in fresh HRT-18 cells at VP10 (Figure 4A), although the titer of BCoV-p95 was lower than that of wt BCoV by 5-fold. To identify whether the sgmRNAs 32K, HE, S, and 12.7, which were undetectable in HRT-18 cells persistently infected with BCoV at 95 d, could be synthesized in fresh HRT-18 cells infected with BCoV-p95, RNA collected at 48 hpi was subject to Northern blot analysis. As shown in Figure 4B, lane 2, sgmRNAs 32K, HE and S were detected. However, sgmRNA 12.7 remained undetectable. The translation efficiency between wt BCoV and BCoV-p95 in fresh HRT-18 cells was also similar, as determined by Western blot analysis (Figure 4C). To further examine whether sgmRNA 12.7 could be synthesized from BCoV-p95, a series of virus passages (VP) were performed in fresh HRT-18 cells (designed VP1-VP10). Consequently, the virus titer of the respective VP3 and VP10 of BCoV-p95 remained at a high level (Figure 4D). However, sgmRNA 12.7 remained undetectable by Northern blotting (Figure 4E). Although sgmRNA 12.7 could be detected by RT-qPCR, its amounts were still very low (Figure 4F). Accordingly, the signals between sgmRNAs E and S observed by Northern blotting (Figure 4B,E, lane 2) were not sgmRNA 12.7 but rather speculated to be unidentified BCoV genomes such as defective interfering RNAs. Regarding the genome structure of BCoV-p95 at VP10 in fresh HRT-18 cells, it was found that (i) the heterogeneous 5′-most sequences disappeared, (ii) the mutation at nt 73 remained, (iii) as with wt BCoV, the 3′-terminal poly(A) tail length varied from 23 to 61 nt at different times of infection and (iv) few synonymous mutations occurred within the ORFs. In addition, the TRS1, which was used for synthesis of sgmRNA 12.7 during persistent infection (Figure 1I), was still employed in regular HRT-18 cells infected with BCoV-p95, suggesting that the structure of sgmRNA 12.7 remained the same at VP10. Together, these results suggest that (i) BCoV-p95 resulting from persistent infection does not increase pathogenicity in comparison with that of wt BCoV in regular HRT-18 cells in the absence of selection pressure and (ii) sgmRNA 12.7 and possibly its encoded protein are not essential for replication in HRT-18 cells. In addition, although the mutation at nt 73 in the genome remained, the results showing that sgmRNAs 32K, HE and S became detectable in fresh HRT-18 cells infected with BCoV-p95 by Northern blot analysis suggest that cellular factors may be responsible for the undetectable sgmRNAs 32k, HE and S in HRT-18 cells persistently infected with wt BCoV. Accordingly, in addition to cellular factors, the remaining undetectable sgmRNA 12.7 may be attributed to the mutation at nt 73.

### 3.5. Evaluation of the Adaptation Capability of BCoV-p95 in Alternate Environments in Comparison to wt BCoV

Although BCoV-p95 can adapt well back to regular host HRT-18 cells in the absence of selection pressure (Figure 4), it remains unknown, in addition to the effect on attenuation of gene expression, whether the altered genome structure in BCoV-p95 affects its fitness when facing the challenges of environmental pressures such as innate immunity and different host cells. To test whether BCoV-p95 could adapt to the environment with innate immune, poly IC-treated HRT-18 cells were respectively infected with wt BCoV and BCoV-p95. The virus (defined as virus passage 0, VP0) was collected at 48 hpi and then passaged in fresh HRT-18 cells (VP1) in the presence of poly IC. At VP10, both viral RNA and proteins were collected and subjected to analyses by Northern blotting (Figure 5A), RT-qPCR (Figure 5B), and Western blotting (Figure 5C). Both viral RNA (Figure 5A, lane 2 and Figure 5B) and proteins (Figure 5C, left panel, lane 4 and right panel) collected from poly IC-treated wt BCoV-infected HRT-18 cells can be detected. In contrast, both viral RNA (Figure 5A, lane 3 and Figure 5B) and proteins (Figure 5C, left panel, lane 5 and right panel) collected from poly IC-treated BCoV-p95-infected HRT-18 cells were not detectable, suggesting that the replication of BCoV-p95 was blocked by the treatment of poly IC-induced innate immunity and thus BCoV-p95 had no adaptation capability in face of the challenges of poly IC-induced innate immunity. To further determine which stage of the BCoV-p95 replication was blocked by the treatment of poly IC-induced innate immunity, a plaque assay and RT-qPCR were performed. As shown in Appendix A, viral replication was almost blocked at VP0 and cannot be detected from VP1 to VP10 based on the results of virus titer (Appendix A) and RT-qPCR (Appendix A), further suggesting that BCoV-p95 had no adaptation capability in face of the challenges of poly IC-induced innate immunity.

Although BCoV-p95 can adapt well back to regular host HRT-18 cells in the absence of selection pressure (Figure 4), it remains unknown, in addition to the effect on attenuation of gene expression, whether the altered genome structure in BCoV-p95 affects its fitness when facing the challenges of environmental pressures such as innate immunity and different host cells. To test whether BCoV-p95 could adapt to the environment with innate immune, poly IC-treated HRT-18 cells were respectively infected with wt BCoV and BCoV-p95. The virus (defined as virus passage 0, VP0) was collected at 48 hpi and then passaged in fresh HRT-18 cells (VP1) in the presence of poly IC. At VP10, both viral RNA and proteins were collected and subjected to analyses by Northern blotting (Figure 5A), RT-qPCR (Figure 5B), and Western blotting (Figure 5C). Both viral RNA (Figure 5A, lane 2 and Figure 5B) and proteins (Figure 5C, left panel, lane 4 and right panel) collected from poly IC-treated wt BCoV-infected HRT-18 cells can be detected. In contrast, both viral RNA (Figure 5A, lane 3 and Figure 5B) and proteins (Figure 5C, left panel, lane 5 and right panel) collected from poly IC-treated BCoV-p95-infected HRT-18 cells were not detectable, suggesting that the replication of BCoV-p95 was blocked by the treatment of poly IC-induced innate immunity and thus BCoV-p95 had no adaptation capability in face of the challenges of poly IC-induced innate immunity. To further determine which stage of the BCoV-p95 replication was blocked by the treatment of poly IC-induced innate immunity, a plaque assay and RT-qPCR were performed. As shown in Appendix A, viral replication was almost blocked at VP0 and cannot be detected from VP1 to VP10 based on the results of virus titer (Appendix A) and RT-qPCR (Appendix A), further suggesting that BCoV-p95 had no adaptation capability in face of the challenges of poly IC-induced innate immunity.

Although BCoV-p95 can adapt well back to regular host HRT-18 cells in the absence of selection pressure (Figure 4), it remains unknown, in addition to the effect on attenuation of gene expression, whether the altered genome structure in BCoV-p95 affects its fitness when facing the challenges of environmental pressures such as innate immunity and different host cells. To test whether BCoV-p95 could adapt to the environment with innate immune, poly IC-treated HRT-18 cells were respectively infected with wt BCoV and BCoV-p95. The virus (defined as virus passage 0, VP0) was collected at 48 hpi and then passaged in fresh HRT-18 cells (VP1) in the presence of poly IC. At VP10, both viral RNA and proteins were collected and subjected to analyses by Northern blotting (Figure 5A), RT-qPCR (Figure 5B), and Western blotting (Figure 5C). Both viral RNA (Figure 5A, lane 2 and Figure 5B) and proteins (Figure 5C, left panel, lane 4 and right panel) collected from poly IC-treated wt BCoV-infected HRT-18 cells can be detected. In contrast, both viral RNA (Figure 5A, lane 3 and Figure 5B) and proteins (Figure 5C, left panel, lane 5 and right panel) collected from poly IC-treated BCoV-p95-infected HRT-18 cells were not detectable, suggesting that the replication of BCoV-p95 was blocked by the treatment of poly IC-induced innate immunity and thus BCoV-p95 had no adaptation capability in face of the challenges of poly IC-induced innate immunity. To further determine which stage of the BCoV-p95 replication was blocked by the treatment of poly IC-induced innate immunity, a plaque assay and RT-qPCR were performed. As shown in Appendix A, viral replication was almost blocked at VP0 and cannot be detected from VP1 to VP10 based on the results of virus titer (Appendix A) and RT-qPCR (Appendix A), further suggesting that BCoV-p95 had no adaptation capability in face of the challenges of poly IC-induced innate immunity.

To determine whether the alterations in genome structure of BCoV-p95 impaired its ability to adapt to other host cells with different tissue origins, mouse L (ML) cells (13), a mouse cell line established from subcutaneous connective tissue, was used. ML cells were respectively infected with wt BCoV and BCoV-p95, and after 24 h of infection, RNA was extracted, and virus was collected to infect fresh ML cells (VP1). The virus passage step was repeated until VP10, and Northern blot analysis was performed to detect the synthesis of viral RNA. For the adaptation of wt BCoV in ML cells, as shown in Figure 5D, viral RNA collected from wt BCoV-infected ML cells was detected, and the amounts of viral RNA in VP2 (lane 3) were less than those in VP10 (lane 4). In addition, the overall amounts of viral RNA at VP2 and VP10 in HRT-18 cells (lanes 1 and 2) were greater than those in ML cells (lanes 3 and 4). However, the amounts of the genome at VP10 in wt-BCoV-infected ML cells were very low in comparison with those of in wt-BCoV-infected HRT-18 cells as detected by RT-qPCR (Figure 5E). Similar results were also observed for synthesis of nsp1 (representing genome translation) at VP10 between wt BCoV-infected HRT-18 cells (Figure 5F, lane 4) and ML cells (Figure 5F, lane 5) in which the signal of nsp1 from wt-BCoV-infected ML cells was relatively low. Although the efficiency of gene expression was low, the results suggest that wt BCoV had been gradually adapting to ML cells. For adaptation of BCoV-p95 in ML cells, in contrast, no detectable viral RNA at VP2 (Figure 5D, lane 9) and VP10 (Figure 5D, lane 10) was found for BCoV-p95 in ML cells by Northern blotting (Figure 5D). Thus, to further determine whether the viral RNA, which was not detectable by Northern blot analysis, were synthesized, RT-qPCR was performed. Although viral RNA for BCoV-p95 in ML cells could be detected by more sensitive RT-qPCR analyses, the amounts of viral RNA of BCoV-p95 were relatively low in comparison with those of wt BCoV in ML cells (~0.7%, ~0.8%, and ~0.2% vs. 100% for genome, S subgenome and N subgenome, respectively). Note that because these values are relatively low, they cannot be shown clearly in Figure 5E. The nsp1 and N protein were not detectable by Western blotting (Figure 5F, lane 8). In addition, although the virus titer can be detected during VP0-VP10 for BCoV-p95 and wt BCoV (Appendix A), the virus titer of BCoV-p95 was lower than that of wt BCoV at VP10 (10^2.8^ pfu/mL for BCoV-p95 and 10^5.9^ pfu/mL for wt BCoV, Appendix A). When the RT-PCR products were subjected to sequencing, sequence analysis revealed that the majority of aa mutations occurred in the S1 subunit of S protein for both wt BCoV (Wt10-ML in Figure 5G) and BCoV-p95 (p9510-ML in Figure 5G) in ML cells at VP10. The results suggest BCoV-p95 can adapt to ML cells, but the adaptation capability is lower in comparison with that of wt BCoV in ML cells.

Together, the results suggest that BCoV-p95 with altered genome structures cannot adapt to the environment with poly IC-induced innate immunity. In addition, although BCoV-p95 can adapt to new host cells (ML cells), the adaptation capability is lower in comparison with that of wt BCoV in ML cells. The results suggest that BCoV-p95 has altered its characteristics even though it can replicate efficiently in regular HRT-18 cells. Since coronavirus nsp16 can lead to reduced recognition of viral RNA by sensor molecules of cell innate immunity [38] and S protein is a main determinant of the cell tropism for coronaviruses [33], we speculated that the main factors leading to the results may be attributed to the altered genome structures of nsp16 and S protein in BCoV-p95 during persistence (Figure 1K,L). In this sense, BCoV-p95 is genetically less robust than wt BCoV.

## 4. Discussion

We in this study employed BCoV, which is in the same genus (*betacoronavirus*) as SARS-COV-2, as a model to understand how coronavirus responded to the selection pressure of persistence by analyzing the alterations of whole genome structure and subsequent virus fitness. The results may have implications for coronavirus gene expression, pathogenicity, and evolution with medical importance and are discussed below.

The high mutation rate of RNA viruses and subsequent alterations of genome structures may be one of the strategies for viruses to rapidly adapt to the new environment in face of selection pressures [39,40,41]. In this study, we have demonstrated that during BCoV persistence the changes in the *cis*-acting elements located in the 5′ and 3′ termini of the genome are at least part of factors leading to the attenuation of gene expression. Specifically, based on the results shown in Figure 2 and Figure 3, it is suggested that the altered 5′ terminal sequences may reduce gene expression efficiency through replication and transcription, but not translation. The reduced poly(A) tail length, on the other hand, may be responsible for the decreased efficiency of replication and translation [28,37]. In addition, the mutation at nt 73 is mainly involved in the sgm RNA synthesis (transcription) but not replication and translation (Figure 3C,E–G). The results fit the general criteria of virus persistence in which viruses adapt to pressure with reduced gene expression to maintain cell physiology for survival.

The striking observation herein was that the amounts of sgmRNAs 12.7, S, 32K and HE, but not N, M and E, in HRT-18 cells during persistence were extremely low in comparison to those in HRT-18 cells at 48 hpi during acute infection (Figure 2B). This gene expression pattern has not been previously identified in coronavirus infection. Based on subsequent analyses (Figure 3 and Figure 4), we argue that the differences between cellular and viral factors in cells during acute and persistent infection are responsible for the outcome because the synthesis of sgmRNAs S, 32K and HE, with the exception of sgmRNA 12.7, was restored in fresh HRT-18 cells infected with BCoV-p95. For sgmRNA 12.7, it has remained puzzling in BCoV why TRS2, which has less sequence complementarity (between the cTRS-L and TRS2-B) than TRS1, is frequently employed as a template switching site for sgmRNA 12.7 synthesis [11,12]. Based on the results shown in Figure 3F,G, it becomes clearly that, in addition to cellular and viral factors in the persistent environment, the mutation at nt 73 within the leader TRS and possibly the secondary structure (SL III) where the leader TRS resides (Figure 1G) are main factors determining the choice of TRS for sgmRNA 12.7 synthesis. These unexpected findings solve the previous unanswered question regarding TRS selection for 12.7 sgmRNA synthesis and thus further extend our knowledge regarding the factors affecting template switching during coronavirus transcription. In addition, the function of 12.7 kDa protein of BCoV has been indicated in counteracting cell innate immune responses in the context of MHV genome [42]. Therefore, concerning why the mutation at nt 73 is selected, it is speculated that the function of 12.7 kDa protein in the cell innate immune response is not required during persistence, and therefore the mutation at nt 73 and TRS1 are selected, resulting in extremely low amounts of sgmRNA 12.7 synthesis. Moreover, this direct evidence also demonstrates that sgmRNA 12.7 is not essential for replication but for virus fitness in the context of BCoV because BCoV-p95 can replicate to a high titer during both persistent and acute infection (Figure 2A and Figure 4A, respectively). Together, this study examining the alterations of genome structure during BCoV persistence unexpectedly solves the previously unanswered questions regarding (i) the coronavirus transcription mechanism of why TRS2 is selected for synthesis of sgmRNA 12.7 during acute wt BCoV infection and (ii) the function of the accessory protein 12.7 kDa, and thus also extends our understanding on the mechanism and factors involving in coronavirus sgmRNA synthesis.

RNA virus has a feature of high mutation rates during RNA replication due to the lack of proofreading capability in its RNA polymerase, resulting in a diverse population of viruses or quasispecies [39,43]. Thus, the genetic structure of RNA virus populations may consist of a network of variants organized in sequence space around a single master sequence [40,44]. Accordingly, regarding how the genome structures of BCoV variants are altered under the environmental challenges such as persistence, immune responses and new hosts in the current study, it is speculated that the mutations in the gene structure of BCoV quasispecies may act as stepping stones toward the synthesis of a new genotype to adapt to a new environment, leading to the alterations of genome structure [41]. Accordingly, we speculate that the observed changes in the viral population occur due to a shift in the quasispecies population. Because the conventional Sanger sequencing was used, the sequences we showed in the passaging experiments of the current study represented the genome structure of the main virus population but not quasispecies. In addition, it has been suggested that populations with mutations of beneficial effect tend to fix early under a sudden environmental change [45]; thus, during the first passage of the infected HRT-18 cells in the persistent infection experiment, whether the survival cells become a source of the persistent cells (founder effect) remains to be determined.

In addition to the discovery of the functional structures of the genome associated with gene expression during persistence, nonsynonymous mutations were also identified and mostly occurred in nsp16 and S protein genes (Figure 1). The aa mutations in S protein occurred mostly in the S1 subunit, the main domain responsible for binding to the cell receptor. For nsp16, in addition to mutations, a 69-aa deletion was consistently identified in persistent infection experiments. Although these mutations were not detrimental for BCoV-p95 replication in regular host HRT-18 cells, the mutations carried fitness costs since the titer for BCoV-p95 was lower by 5-fold compared with that of wt BCoV in regular host HRT-18 cells (Figure 4A). Therefore, the selected BCoV-p95 through persistence with fitness cost does not have increased pathogenicity in regular host HRT-18 cells and thus extend our knowledge on coronavirus infection. Since the selected variant may not increase its pathogenicity, the results may contribute to the disease control of coronavirus. Whether the co-infection of BCoV-p95 and wt BCoV in HRT-18 cells leads to increased pathogenicity and whether wt BCoV in HRT-18 can outcompete with BCoV-p95 are also important questions required to be further determined.

However, the ability of wt BCoV and BCoV-p95 to replicate efficiently in fresh HRT-18 cells raises the question of why wt BCoV but not BCoV-p95 can withstand the selection pressure of innate immunity (in poly IC-treated HRT-18 cells, Figure 5A–C) and rapidly adapt to new host cells (ML cells) (Figure 5D–F). Since coronavirus nsp16 can lead to reduced recognition of viral RNA by sensor molecules of cell innate immunity [38], we speculated that the main factors leading to inability in adapting to the environment with innate immunity may be attributed to the altered genome structures in nsp16 of BCoV-p95 during persistence. Regarding the question of the capability in adapting different host cells between wt BCoV and BCoV-p95, we reason this as follows. It has been suggested that RNA viruses with genetic robustness allow the viral population to explore an extensive region of sequence space, leading to numerous individuals that can rapidly withstand the environmental changes [46]. We therefore reason that the populations in wt BCoV can develop variants with extensive sequence space and then rapidly synthesize new genotypes for adaptation to new environments. However, the variants developed by BCoV-p95 may occupy less sequence space and thus may take a longer time to acquire the new genotype for adaptation. This argument in turn suggests that wt BCoV is genetically more robust than BCoV-p95 and is consistent with the finding that wt poliovirus occupies a sequence space that enables it to rapidly adapt to environmental pressure [47].

## 5. Conclusions

In conclusion, the selection pressure of BCoV persistence leads to alterations of genome structure, attenuation of gene expression and the synthesis of subgenomic mRNA (sgmRNA) with a previously unidentified pattern in persistently infected HRT-18 cells. The altered genome structures are associated with the attenuation of gene expression during persistence. In addition, the genome structure at the 5′ terminus and cellular environment are responsible for the sgmRNA synthesis, solving the previously unanswered question regarding the selection of TRS for synthesis of BCoV sgmRNA 12.7 and thus further extending our knowledge of the coronavirus transcription mechanism. Although the resultant BCoV (BVoC-p95) still replicates with high efficiency in regular host HRT-18 cells, it still displays reduced gene expression and pathogenicity in comparison with those of wt BCoV. In addition, when compared with wt BCoV, the BCoV-p95 cannot rapidly adapt to the environment with innate immune and to other host cells with different tissue origins, suggesting that the selected BCoV-p95 has altered its characteristics. Although different selection pressures may lead to different outcomes, these previously unidentified features may provide important information contributing to disease control and clinical treatments against coronavirus infection including SARS-CoV-2.

## Figures and Tables

**Figure 1 cells-09-02322-f001:**
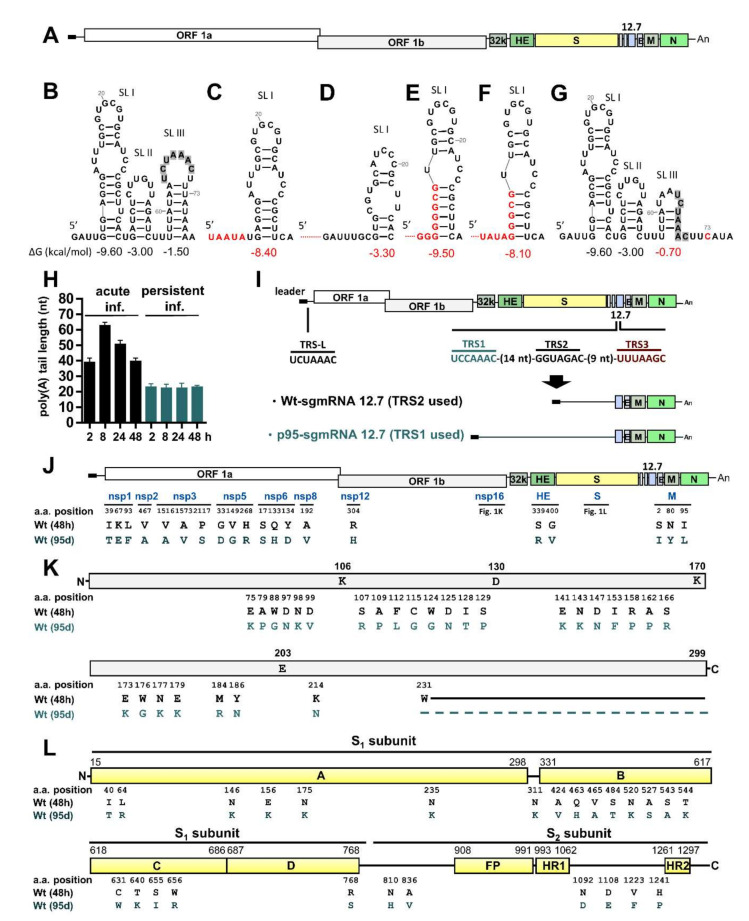
Alterations of BCoV genome structures under persistent infection. (**A**) Schematic diagram of the BCoV genome structure. (**B**) *Cis*-acting RNA elements SLs I-III at the 5′-proximal region of the BCoV genome. The core sequences of the leader (CS-L; UCUAAAC) in the transcription regulatory sequence (TRS) are shaded in gray. ΔG: free energy. (**C**–**F**) Altered SL I structure caused by nt mutations (indicated in red) at the 5′ terminus of the BCoV genome during persistent infection. The nt deletion is indicated with a red dot. (**G**) Altered SL III structure due to mutation at nt 73 (indicated in red). The core sequences of the leader (CS-L; UCUAAAC) in the TRS are shaded in gray. (**H**) Poly(A) tail length at the 3′ terminus of the BCoV genome at different times of acute and persistent infection. Inf.: infection. (**I**) Upper panel: Illustration of core sequences of leader TRS (TRS-L) and body TRSs (TRS-B) TRS1, TRS2 and TRS3 in the BCoV genome employed for the synthesis of sgmRNA 12.7. Lower panel: The structure of the synthesized sgmRNA 12.7 with TRS2 (Wt-sgmRNA 12.7) or TRS1 (p95-sgmRNA 12.7). (**J**) Location of the mutated aa in the BCoV genome (except for nsp16 and S protein) during persistence. Wt (48 h): Viral RNA collected from fresh HRT-18 infected with wt BCoV at 48 hpi. Wt (95 d): Viral RNA collected from HRT-18 cells after 95 d of persistent infection with wt BCoV. (**K**) Linear schematic of nsp16 showing the positions of the conserved catalytic tetrad (K-D-K-E) and identified aa alterations during persistence. The aa deletion is indicated with a dash. (**L**) Linear schematic of BCoV S protein with subunits S1 (domains A–D) and S2 (FP, HR1 and HR2) showing the positions of the aa alterations identified during persistence. The residue numbering is based on the BCoV Mebus strain spike protein (GenBank: U00735) with domain boundaries based on the HCoV-OC43 S structure [33]. 32K: 32 kDa protein, HE: hemagglutinin/esterase, S: spike protein, 12.7: 12.7 kDa protein, E: envelope protein, M: membrane protein, N: nucleocapsid protein, FP: fusion peptide, HR1: heptad repeat 1, HR2: heptad repeat 2.

**Figure 2 cells-09-02322-f002:**
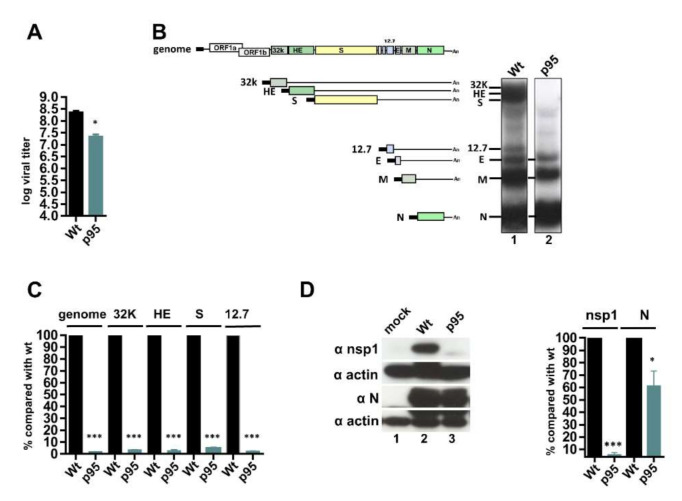
Determination of gene expression efficiency for BCoV during persistence. (**A**) The virus titer of wt BCoV (Wt) and BCoV-p95 (p95) as determined by the plaque assay. Virus collected from fresh HRT-18 infected with wt BCoV was designated Wt. Virus collected from HRT-18 cells after 95 d of persistent infection with wt BCoV was designated BCoV-p95 (p95). (**B**) Left panel: Schematic representing the BCoV genome and sgmRNAs. Right panel: Synthesis of BCoV RNA as determined by Northern blotting. (**C**) Relative amounts of genome and subgenomes including sgmRNAs 32K, HE, S and 12.7 between Wt and p95 as measured by RT-qPCR. (**D**) Left panel: Coronaviral protein synthesis from the genome (represented by nsp1) and subgenome (represented by N protein) during persistence by Western blotting. Right panel: Relative amounts of nsp1 and N protein between Wt and p95 based on the results shown in the left panel. The values in (**A**,**C**,**D**) represent the mean ± standard deviation (SD) of three individual experiments. Statistical significance was evaluated using a *t*-test: * *p* < 0.05, *** *p* < 0.001.

**Figure 3 cells-09-02322-f003:**
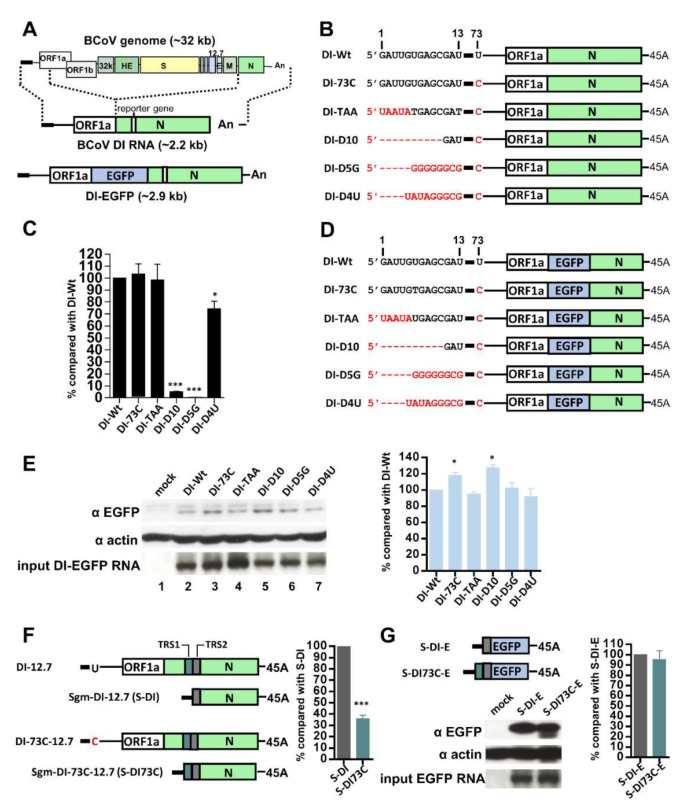
Effects of alterations in the 5′ and 3′ termini of the genome and subgenomes caused by persistence on gene expression. (**A**) Structure of naturally occurring BCoV DI RNA related to the BCoV full-length genome. DI RNA was used for the replication assay, and its derivative DI-EGFP containing EGFP gene was employed for the translation assay. (**B**) DI RNA constructs with 5′-mutated sequences (Figure 1C–G) identified during persistence. (**C**) Relative levels of DI RNA synthesis between DI RNA constructs illustrated in (**B**) as measured by RT-qPCR. (**D**) DI-EGFP constructs with 5′-mutated sequences (Figure 1C–G) identified during persistence. (**E**) Left panel: Determination of the synthesis of DI-EGFP fusion protein from the constructs illustrated in (**D**) by Western blotting. Right panel: Relative amounts of DI-EGFP fusion protein between DI-EGFP constructs based on the results shown in the left panel. (**F**) Left panel: Genome structure of DI-12.7 and DI-73C-12.7 with insertions of TRS1 (green), TRS2 (gray) and TRS3 (not shown in color) for the 12.7 gene, and predicted sgm DI RNAs (Sgm-DI-12.7 and Sgm-DI-73C-12.7, respectively) derived from the aforementioned DI RNA constructs. Right panel: Relative amounts of sgm DI RNA synthesis between DI RNA 12.7 and DI-73C-12.7, as quantitated by RT-qPCR. (**G**) Left upper panel: Constructs of the EGFP gene-containing sgm DI RNA S-DI-E (synthesized with TRS2) and S-DI73C-E (synthesized with TRS1). Left lower panel: Determination of the synthesis of EGFP from constructs illustrated in the left upper panel by Western blotting. Right panel: Relative amounts of EGFP between sgm DI RNA constructs based on the results shown in the left lower panel. The values in (**C**) and (**E**–**G**) represent the mean ± standard deviation (SD) of three individual experiments. Statistical significance was evaluated using a *t*-test: * *p* < 0.05, *** *p* < 0.001.

**Figure 4 cells-09-02322-f004:**
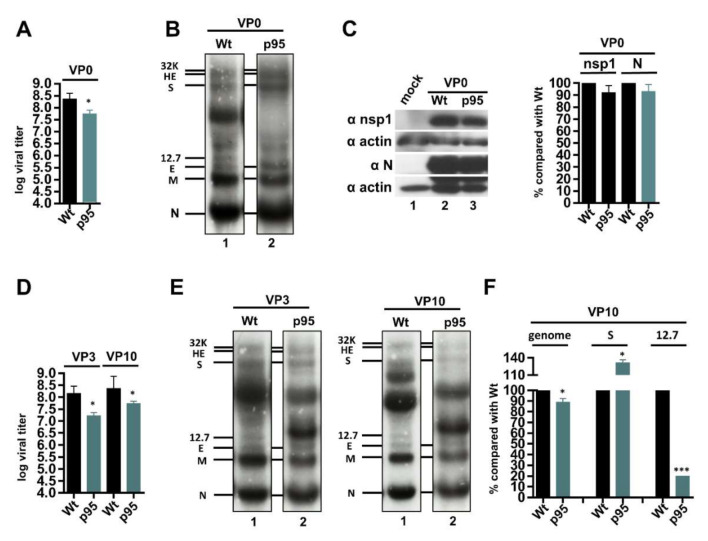
Characterization of the gene expression of BCoV-p95 in regular host HRT-18 cells. (**A**) The virus titer of wt BCoV (Wt) and BCoV-p95 (p95) in fresh HRT-18 cells at VP0 as determined by the plaque assay. (**B**) Determination of BCoV RNA synthesis from fresh HRT-18 cells infected with Wt or p95 at VP0 by Northern blotting. (**C**) Left panel: Coronavirus protein synthesis from the genome (represented by nsp1) and subgenome (represented by N protein) from fresh HRT-18 cells infected with Wt or p95 at VP0 by Western blotting. Right panel: Relative amounts of nsp1 and N protein between Wt and p95 in fresh HRT-18 cells at VP0 based on the results shown in left panel. (**D**) The virus titer of Wt and p95 in fresh HRT-18 cells at VP3 and VP10 as determined by the plaque assay. (**E**) Determination of BCoV RNA synthesis from fresh HRT-18 cells infected with Wt or p95 at VP3 (left panel) and VP10 (right panel) by Northern blotting. (**F**) Relative amounts of genome and sgmRNAs (represented by S and 12.7) between Wt and p95 from fresh HRT-18 cells infected with Wt or p95 at VP10 as measured by RT-qPCR. The values in (**A**), (**C**), (**D**) and (**F**) represent the mean ± standard deviation (SD) of three individual experiments. Statistical significance was evaluated using a *t*-test: * *p* < 0.05, *** *p* < 0.001. VP: virus passage.

**Figure 5 cells-09-02322-f005:**
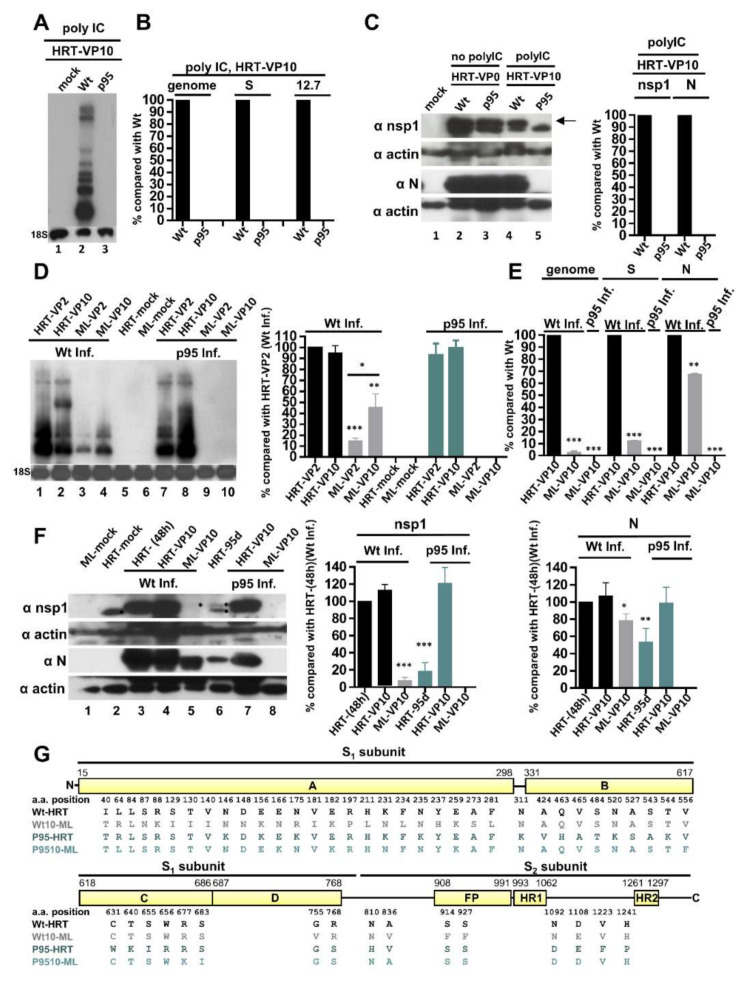
Evaluation of the adaptation capability of BCoV-p95 in poly IC-treated HRT-18 cells and ML cells. (**A**) Determination of BCoV RNA synthesis from fresh HRT-18 cells infected with Wt or p95 in the presence of poly IC at VP10 by Northern blotting. (**B**) The relative amounts of genome and sgmRNAs (represented by sgmRNAs S and 12.7) between Wt and p95 from fresh HRT-18 cells infected with Wt or p95 in the presence of poly IC at VP10 as measured by RT-qPCR. (**C**) Left panel: Coronavirus protein synthesis from the genome (represented by nsp1) and subgenome (represented by N protein) from fresh HRT-18 cells infected with Wt or p95 in the presence or absence of poly IC at VP0 and VP10 by Western blotting. Right panel: Relative amounts of nsp1 and N protein between Wt and p95 based on the results shown in the left panel. (**D**) Left panel: Determination of BCoV RNA synthesis from fresh HRT-18 or ML cells infected with Wt or p95 at VP2 or VP10 by Northern blotting. Right panel: Relative amounts of Wt and p95 BCoV RNA based on the results shown in the left panel. (**E**) The relative amounts of genome and sgmRNAs (represented by sgmRNAs S and 12.7) from fresh ML cells infected with Wt or p95 at VP10 as measured by RT-qPCR. (**F**) Left panel: Coronavirus protein synthesis from the genome (represented by nsp1) and subgenome (represented by N protein) from fresh HRT-18 or ML cells infected with Wt or p95 at VP10 by Western blotting. Middle and right panel: Relative amounts of nsp1 and N protein, respectively, between Wt and p95 based on the results shown in the left panel. The asterisk in lane 5 indicates the relatively weak signal of nsp1. The black dot indicates the background signal of mock-infected HRT-18 cells. (**G**) Linear schematic of BCoV S protein showing the comparison of the aa sequences identified from HRT-18 cells infected with Wt (Wt-HRT), ML cells infected with Wt at VP10 (Wt10-ML), HRT-18 cells persistently infected with p95 (p95-HRT) at 95 d and ML cells infected with p95 at VP10 (p9510-ML). The values in (**B**–**F**) represent the mean ± standard deviation (SD) of three individual experiments. Statistical significance was evaluated using a *t*-test: * *p* < 0.05, ** *p* < 0.01, *** *p* < 0.001. VP: virus passage, HRT: HRT-18 cells, ML cells: mouse L cells, Inf.: infection, FP: fusion peptide, HR1: heptad repeat 1, HR2: heptad repeat 2.

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
