# Peer review of "Effects of Coronavirus Persistence on the Genome Structure and Subsequent Gene Expression, Pathogenicity and Adaptation Capability"

_cells, 2020, doi:10.3390/cells9102322_

Round 1
Reviewer 1 Report
In this manuscript, the authors studied the effect of persistent infection of bovine coronaviruses and observed alterations of genome structure as well as gene expression. The results are clearly described and address an important topic. Even though reports of persistent infection with the highly pathogenic coronaviruses MERS-CoV, SARS-CoV, and SARS-CoV-2 are scarce, it is important to understand the molecular mechanisms of coronavirus replication that can also lead or contribute to persistent infection thereby contributing to the pathogenesis. However, there are some issue that should be addressed.
It seems that for the sequencing conventional sanger sequencing was used and not NGS. Using this method, the authors are not able to detect the viral quasispecies. Especially in the context of passaging experiments it should be addressed if the observed changes in the viral population occur due to de novo mutations or a shift in the quasispecies population.
In Figure 4 and 5, the authors claim that BCoV-p95 is more susceptible towards immune pressure and is not able to adapt to new host cells. Plaque assays should be performed to show the effect on viral titer. Along the same line, how did the authors normalize their input when comparing passaged and non-passaged viruses? Was a specific MOI used? The input should always be normalized, to draw conclusions about replication fitness and susceptibility to immune pressure.
No information about the passaging experiment itself e.g. differences during the passaging between passaged virus and WT with respect to viral titers is being shown. This should be included.
In order to claim that the observed changes including the unique pattern of sgmRNA synthesis, are representative for persistent infection, 3 biological replicates should have been included, meaning that 3 viruses should have been passaged over the whole time. However, I do not insist on repeating the whole experiment if that has not been included from the beginning.
Minor points:
The difference between Figure 1A and Figure 3A is not quite clear, please specify.
Minor spell check should be performed (eg. lines 459, 484, 534)
Author Response
Dear Reviewer,
We thank you for the detailed reviews and the valuable comments for improvement of our manuscript. We have responded to the comments with details, in particular experimental design and their interpretation presented in the previous manuscript, as follows.
Reviewer #1
In this manuscript, the authors studied the effect of persistent infection of bovine coronaviruses and observed alterations of genome structure as well as gene expression. The results are clearly described and address an important topic. Even though reports of persistent infection with the highly pathogenic coronaviruses MERS-CoV, SARS-CoV, and SARS-CoV-2 are scarce, it is important to understand the molecular mechanisms of coronavirus replication that can also lead or contribute to persistent infection thereby contributing to the pathogenesis. However, there are some issue that should be addressed.
1. It seems that for the sequencing conventional the Sanger sequencing was used and not NGS. Using this method, the authors are not able to detect the viral quasispecies. Especially in the context of passaging experiments it should be addressed if the observed changes in the viral population occur due to de novo mutations or a shift in the quasispecies population.
Authors’ response:
We thank the reviewer for the valuable suggestion. We have addressed this according to the reviewer’s suggestion by stating “because the conventional Sanger sequencing was used, the sequences shown in the passaging experiments represented the genome structure of the main virus population but not the quasispecies. We also speculate that the observed changes in the viral population may occur due to a shift in the quasispecies population in the Discussion section. These have been addressed on page 3, lines 136-138; page 16, lines 536-547.
2. In Figure 4 and 5, the authors claim that BCoV-p95 is more susceptible towards immune pressure and is not able to adapt to new host cells. Plaque assays should be performed to show the effect on viral titer. Along the same line, how did the authors normalize their input when comparing passaged and non-passaged viruses? Was a specific MOI used? The input should always be normalized, to draw conclusions about replication fitness and susceptibility to immune pressure.
Authors’ response:
For the infection experiment of BCoV-p95 back in fresh HRT-18 cells in Figure 4, specific MOI (1 MOI) was used for each passage for both BCoV-p95 and wt BCoV. This has been addressed in page 3, lines 108-110.
For the adaptation experiment of BCoV-p95 in poly IC-induced innate immunity in Figure 5, we thank the reviewer for the suggestion to clarify the experimental design in order to draw the conclusion. We used specific MOI and therefore, for the first time of infection, 1 MOI was used for both BCoV-p95 and wt BCoV. The titer was 107.7.pfu/mL for wt BCoV; however, almost no virus plaque can be detected for BCoV-p95 (101.4 pfu/mL) at VP0. In order to determine whether it can gradually adapt to the challenge, we performed a blind passage until VP10 and found it cannot replicate since VP1 (see virus titer results in Figure S4A in the revised manuscript). The virus titer at VP10 is also included in Figure S4B. These have been addressed on page 3, lines 110-118 and Figure S4.
Since there is no replication, there is no adaptation. We therefore modified the conclusion as follows: “the replication of BCoV-p95 was blocked by the treatment of poly IC-induced innate immunity” and “BCoV-p95 had no adaptation capability in face of the challenges of poly IC-induced innate immunity”. These have been addressed on page 12, lines 415-417.
For the adaptation experiment of BCoV-p95 in alternative host cells (ML cells), we thank the reviewer for pointing out the unclear description. we used specific MOI and therefore, for the first time of infection, 1 MOI was used for both BCoV-p95 and wt BCoV. The titer was 105.2 pfu/mL for wt BCoV and 103.8 pfu/mL for BCoV-p95 at VP0. Therefore, for VP1-VP10 infection, 0.001 MOI was used for both viruses. The virus titer at VP10 is included in Figure S6B. These have been addressed on page 3, lines 119-123.
In addition, we in the original manuscript on page 12, lines 416-417, have described that BCoV-95 can replicate although the replication efficiency was relatively low in comparison with that of wt BCoV at VP10. Because BCoV-p95 genomic RNA in ML cells can be detected by RT-PCR, we then can sequence the S protein gene at VP10 as shown in the original manuscript (p9510-ML in figure 5G). Furthermore, because the replication efficiency of BCoV-p95 genomic and subgenomic RNA in ML cells was relatively low in comparison with that of wt BCoV (~0.7%, ~0.8% and ~0.2% vs 100% for genome, S subgenome and N subgenome, respectively) in ML cells, consequently, this very low number cannot be shown clearly in the Figure E within a percentage (%) scale. This may cause misunderstanding that BCoV-95 cannot replicate in ML cells. Therefore, based on the results of sequencing analysis from detected genomic RNA by RT-PCR (Figure 5G), and virus titer at VP0-VP10 (Figure S6), it is concluded that BCoV-p95 can replicate and then can adapt to ML cells, but the adaptation capability is lower than that of wt BCoV. To clarify the description and to show it can replicate, we have added the relative value (~0.7%, ~0.8% and ~0.2% vs 100%) in the description of the revised manuscript and included the detected virus titer in Figure S6. These have been addressed on page 12, lines 440-450; page 13, lines 451-457 and Figure S6.
3. No information about the passaging experiment itself e.g. differences during the passaging between passaged virus and WT with respect to viral titers is being shown. This should be included.
Authors’ response:
We thank the reviewer for the valuable suggestion to improve our manuscript and we have included the virus tires at VP0-VP10 in Figures S3, S4 and S6, and addressed these on page 10, lines 355-356; page 12, lines 419-422; page 12, lines 446-449.
4. In order to claim that the observed changes including the unique pattern of sgmRNA synthesis, are representative for persistent infection, 3 biological replicates should have been included, meaning that 3 viruses should have been passaged over the whole time. However, I do not insist on repeating the whole experiment if that has not been included from the beginning.
Authors’ response:
We thank the reviewer for the valuable suggestion although we did not perform the experiments. However, it would be more appropriate to delete the “unique” and use “previously unidentified” to describe the changed pattern of subgenomic mRNA expression. We have addressed this on page 1, line 22; page 17, line 586.
Minor points:
1. The difference between Figure 1A and Figure 3A is not quite clear, please specify.
Authors’ response:
We have modified the two figures (Figures 1A and 3A) to be consistent by using the same genome structure and colors. The Figure 1A simply represents ~32 Kb of BCoV full-length genome. However, Figure 3A shows the structure of the ~32 Kb of BCoV full-length genome (upper panel) related to its naturally occurring BCoV defective interfering (DI) RNA (~2.2 Kb) and its derivative DI-EGFP (~2.9 Kb) (lower two panels). DI RNA was used for the replication assay, and its derivative DI-EGFP containing EGFP gene was employed for the translation assay. BCoV DI RNA is a surrogate for the BCoV genome and has been extensively used to study cis-acting RNA elements required for coronavirus gene expression.
2. Minor spell check should be performed (eg. lines 459, 484, 534)
Authors’ response:
We thank the reviewer for pointing out the incorrect spelling. We have made the corrections (adaption to adaptation; expressionn to expression; comaprison to comparison; diaplay to display) in the revised manuscript on page 3, line 104; page 8, line 290; page 8, line 300; page 8, line 305.
Sincerely,
Hung-Yi Wu, D.V.M., M.S., Ph.D.
Professor of Graduate Institute of Veterinary Pathobiology
College of Veterinary Medicine
National Chung Hsing University
Tel:886-4-22840369
Fax:886-4-22862073
Email:hwu2@dragon.nchu.edu.tw

Reviewer 2 Report
This scientific work is of interest to virologists and molecular biologists, contains important new scientific information. No comments and I recommend it for publication.
Author Response
Dear Reviewer,
We thank you for the effort to review the manuscript.
Reviewer #2
This scientific work is of interest to virologists and molecular biologists, contains important new scientific information. No comments and I recommend it for publication.
Sincerely,
Hung-Yi Wu, D.V.M., M.S., Ph.D.
Professor of Graduate Institute of Veterinary Pathobiology
College of Veterinary Medicine
National Chung Hsing University
Tel:886-4-22840369
Fax:886-4-22862073
Email:hwu2@dragon.nchu.edu.tw

Reviewer 3 Report
Effects of coronavirus persistence on the genome structure and subsequent gene expression, pathogenicity and adaptation capability
Ching-Hung Lin1, Cheng-Yao Yang1, Meilin Wang2, Shan-Chia Ou3, Chen-Yu Lo1, Tsung-Lin 5 Tsai1, Hung-Yi Wu1*
In this study, the authors show that bovine coronavirus can establish persistent infection in a small percentage of HTR-18 cells. After passing these cells 95 times, they analyze viral RNA and viral protein expression in these cells and show changes in the structure of the viral genome associated with decreased viral protein expression. This is followed by a characterization of this stock (BCoV-p95) of so-called persistent virus.
It is an exciting topic and a clearly interesting study of great topicality; however, I have some problems with the understanding of this article, I lack information for the full understanding of the results and I also partly disagree on the interpretation of some results on the basis of the results shown.
From a general point of view, for a non-specialist of Coronaviruses, the manuscript seems sometimes reserved for specialists (for a non afficionados to know that mouse hepatitis virus is a coronavirus is not intuitive, the nomenclature of TRS is not friendly, these are only examples). In addition, sometimes there is missing information to fully understand how the experiments are performed. This makes the ms difficult to read even though the content is exciting.
Fig. 1 and 2
The results presented are really nice and interesting
Just to be sure, we are talking about RNA coming from the persistently infected cell? not RNA coming from the budding virus in the supernatant?
There is some information I would have liked to have about what happens during these 95 passages. All the cells are infected all the time? what about the growth of these cells vs. uninfected cells? what about a possible cytopathic effect or cell death? what happens to the virus produced in the supernatant? is it likely to re-infect persistently infected cells (or are these cells refractory to superinfection?)? in other words can these cells be superinfected with BCoV wt during the passages?
In addition, there seems to be a large bias introduced at the time of the first infection. If I understand correctly for the persistence experiment, cells were infected at a MOI of 1 (one infectious particle per cell), less than 5% (?) of the cells survive and are the source of the persistent cells, this may represent a bias (founder effect), is there a way to analyze the viral sequences in those surviving cells from the first infection.
Fig1D not easy to understand where is the mutation that induces this conformational change.
Fig 1.I Not sure I understand the nomenclature about TRS1, 2 ,3
Fig2 again it is not necessarily clear to me what we are talking about, according to my understanding what is shown in fig 2 B,C,D is the situation inside persistent cells.
My question here is whether there is a difference between what is found in the cells and what is budding? In other words is there a bias in the budding that could explain the relatively small difference in the titer of the virus from passage 95 vs. the wt virus.
Is it possible to sequence the RNA found in the supernatant versus the intracellular RNA?
From my point of view this is an important information to fully understand and interpret the results presented in Figure 4.
I would also be curious to know what happens in an HRT-18 cell co-infection experiment (BCoV-wt and -p95), whether one of the viruses outcompete the other and how fast.
I have to admit that my main problem comes from the results presented in Figure 5.
The topic is the ability of the BCoV-p95 stock to adapt to a new environment. This is such an exciting and important topic, starting from a virus that is supposed to have adapted to such a particular environment. While the project is interesting, I don't think the results presented in this figure meet this objective. From my point of view they face a serious pitfall, in order for the quasi-species to work and for evolution to take place, there must be replication, if there is no replication, there is no mutation, there is nothing to select and therefore no possible evolution, and I am afraid that in the two models presented there is no replication of BCoV-p95 so...
In the model with poly-I:C why only present the situation in passage 10, I have the impression here that what is addressed is the ability of both viruses to escape an antiviral state, if the BCoV-p95 is so sensitive to the antiviral state induced by poly-I:C and its replication is blocked it simply can't evolve and what is measured here is the sensitivity to the antiviral state which may be interesting but has nothing to do with the ability to evolve. Show us what happens in the early passages.
In the mouse cell model it is pretty much the same, clearly the design of the experiment is interesting and related to the problematic of evolution but if there is no replication of the BCoV-p95 in the mouse cells it simply cannot have evolution and there is no evidence that the BCoV-p95 replicates in these mouse cells while the wt. virus clearly replicates in those cells and has a chance to adapt to this new environment.
Author Response
Dear Reviewer,
We thank you for the detailed reviews and the valuable comments for improvement of our manuscript. We have responded to the comments with details, in particular the experimental information, and make the manuscript more readable as follows.
Reviewer #3
In this study, the authors show that bovine coronavirus can establish persistent infection in a small percentage of HTR-18 cells. After passing these cells 95 times, they analyze viral RNA and viral protein expression in these cells and show changes in the structure of the viral genome associated with decreased viral protein expression. This is followed by a characterization of this stock (BCoV-p95) of so-called persistent virus.
It is an exciting topic and a clearly interesting study of great topicality; however, I have some problems with the understanding of this article, I lack information for the full understanding of the results and I also partly disagree on the interpretation of some results on the basis of the results shown.
From a general point of view, for a non-specialist of Coronaviruses, the manuscript seems sometimes reserved for specialists (for a non afficionados to know that mouse hepatitis virus is a coronavirus is not intuitive, the nomenclature of TRS is not friendly, these are only examples). In addition, sometimes there is missing information to fully understand how the experiments are performed. This makes the ms difficult to read even though the content is exciting.
Fig. 1 and 2
1.The results presented are really nice and interesting.
Just to be sure, we are talking about RNA coming from the persistently infected cell? not RNA coming from the budding virus in the supernatant?
Authors’ response:
In coronavirus, in addition to replication of genomic RNA (genome), a nested set of subgenomic mRNAs (sgmRNAs) are synthesized using the genome as a template in infected cells. However, due to the lack of packaging signal in subgenomic mRNAs, the budding virus particles only contain genomic RNA (genome) (see the reference below). Accordingly, the budding virus particles contain only virus genome; however, the infected cells contain both virus genome and 7 subgenomes. Yes, the detected RNAs (including genomic RNA and subgenomic RNA) are from persistently infected cells. We have addressed this on page 2, lines 49-52 in the revised manuscript.
Reference:
1.Masters, P. S., Coronavirus genomic RNA packaging. Virology 2019, 537, 198-207.
2. There is some information I would have liked to have about what happens during these 95 passages. All the cells are infected all the time? what about the growth of these cells vs. uninfected cells? what about a possible cytopathic effect or cell death? what happens to the virus produced in the supernatant? is it likely to re-infect persistently infected cells (or are these cells refractory to superinfection?)? in other words can these cells be superinfected with BCoV wt during the passages?
Authors’ response:
During the persistence, cytoplasmic vacuolization and floating cells can be observed. Cytopathic effect can be found occasionally but is not much. Although the shape of the cells is similar between the uninfected and persistently infected cells, the persistently infected cells grow slower. In addition, we also performed superinfection and the results suggest that the persistently infected cells cannot be infected with wt BCoV based on the results of (i) sequencing and (ii) gene expression in which the RNA and protein were not increased with the time of superinfection. We did not test whether all the cells are infected all the time; however, based on the superinfection results, it is speculated that the cells may be almost infected.
3. In addition, there seems to be a large bias introduced at the time of the first infection. If I understand correctly for the persistence experiment, cells were infected at a MOI of 1 (one infectious particle per cell), less than 5% (?) of the cells survive and are the source of the persistent cells, this may represent a bias (founder effect), is there a way to analyze the viral sequences in those surviving cells from the first infection.
Authors’ response:
We agree that the surviving cells after first passage could be the source of the persistent cells because it has been suggested that under a sudden environmental change, mutations with beneficial effect tended to fix early, followed by mutations of smaller beneficial effect. Consequently, analyzing the viral sequences in those surviving cells from the first passage is a way to determine the hypothesis. Although we did not perform the experiment, we have addressed this on Discussion section on page 16, lines 547-551.
4. Fig1D not easy to understand where is the mutation that induces this conformational change.
Authors’ response:
We thank the reviewer for pointing out the unclear description. The 5’ first 10 nucleotides (5’−−−−−−−−−−GAU…)of the genome shown in Fig. 1D are deleted (indicated by red dashes) in comparison with those (5’GAUUGUGAGCGAU…) shown in the 5’ Fig. 1B (wt BCoV, the 5’ first 10 nucleotides are not deleted) and that is the reason leading to the conformational change. The 5’ sequence difference between the two can also be clearly seen in Figure 3B DI-Wt and DI-D10. We have clarified this on page 6, lines 232-233 and Figure 1D.
5. Fig 1.I Not sure I understand the nomenclature about TRS1, 2 ,3
Authors’ response:
We thank the reviewer for pointing out the unclear description. We understand transcription regulatory sequence (TRS) is really not friendly because coronavirus transcription (synthesis of subgenoms) is a very complicated process. In words, TRS is a key sequence motif to synthesize coronavirus subgenome by similarity-assisted homologous recombination mechanism. We have explained the structure, location and the function of coronavirus TRS, and how TRS functions to synthesize subgenomic mRNA (subgenome or sgmRNA) on page 2, lines 49-60. In most of the cases, only one TRS located upstream of each structural gene is identified; however, BCoV 12.7 kDa gene contains three TRSs (designated TRS1, TRS2 and TRS3) located upstream of 12.7 kDa gene (Fig. 1I). Because TRS1, TRS2 and TRS3 all show different levels of complementarity to the 5’ leader TRS (cTRS-L), they all can be potentially used to synthesized subgenomic mRNA 12.7 kDa (sgmRNA 12.7) through base pairing by similarity-assisted homologous recombination mechanism in infected cells. These have been addressed in page 2, lines 60-68.
Together, we have modified the Figure 1I to increase the clarity and explained the TRS in page 2, lines 49-68; page 6, lines 236-237 and Figure 1I.
6. Fig2 again it is not necessarily clear to me what we are talking about, according to my understanding what is shown in fig 2 B,C,D is the situation inside persistent cells.
My question here is whether there is a difference between what is found in the cells and what is budding? In other words is there a bias in the budding that could explain the relatively small difference in the titer of the virus from passage 95 vs. the wt virus.
Is it possible to sequence the RNA found in the supernatant versus the intracellular RNA?
From my point of view this is an important information to fully understand and interpret the results presented in Figure 4.
Authors’ response:
Figure 2
It has been known in coronavirus that (i) the budding virus particle contain only virus genome; however, the infected cells contain both virus genome and 7 subgenomes (see figure 2, left panel) and that (ii) the number ratio of genome vs 7 species of subgenomes is 1 vs 10-300 in infected cells (different subgenomes have different number of RNA molecules in infected cells). Therefore, what is found in virus particle is different from in infected cells. The aim of Fig 2 is to compare the efficiency of gene expression (replication and translation) between passage 95 (BCoV-p95) in the persistent cells. vs. the wt BCoV in the fresh cells. Thus, to represent the efficiency of gene expression (replication and translation) for coronavirus, (i) the number of virus particle, (ii) the amounts of RNAs including genome and sungenomes in infected cells and (iii) protein synthesis from genome and subgenomes are all required to be considered. In addition, the budding virus particle contains only virus genome. Consequently, there could be a bias if we only detect virus titer (the number of genome-containing budding virus particle) to represent replication. Thus, this may also possibly explain the relatively small difference in the titer of the virus from passage 95 vs. the wt virus in terms of the value of log (7.34 vs 8.35). That is the reason why we also have to detect the synthesis of genome RNA and subgenome RNA in infected cells by Northern analysis (Fig. 2B) and RT-qPCR (Fig. 2C) to represent replication efficiency as well as protein by Western blot (Fig. 2D) to represent translation efficiency. These have been addressed on page 7, lines 251-252, lines 254-256 and lines 260-262; page 8, lines 279-280, lines 282-283.
Figure 4
We thank the reviewer for the suggestion to improve the clarity in interpreting Figure 4, and yes, before we performed the experiment for Figure 4, we have done the sequence analysis for the supernatant to ensure the genome structure from the supernatant viruses is not altered in comparison with the genome structure from persistently infected cells. Thus, we then can exclude the possibility of altered sequence structure affecting the interpretation of the results. We have addressed this to increase the clarity on page 10, lines 351-354.
7. I would also be curious to know what happens in an HRT-18 cell co-infection experiment (BCoV-wt and -p95), whether one of the viruses outcompete the other and how fast.
Authors’ response:
We thank the reviewer for the suggestion. We did not perform the experiment. However, in terms of evolution, it would be an interesting experiment to perform and we have addressed this in the Discussion section on page 16, lines 562-565.
8. I have to admit that my main problem comes from the results presented in Figure 5.
The topic is the ability of the BCoV-p95 stock to adapt to a new environment. This is such an exciting and important topic, starting from a virus that is supposed to have adapted to such a particular environment. While the project is interesting, I don't think the results presented in this figure meet this objective. From my point of view they face a serious pitfall, in order for the quasi-species to work and for evolution to take place, there must be replication, if there is no replication, there is no mutation, there is nothing to select and therefore no possible evolution, and I am afraid that in the two models presented there is no replication of BCoV-p95 so...
In the model with poly-I:C why only present the situation in passage 10, I have the impression here that what is addressed is the ability of both viruses to escape an antiviral state, if the BCoV-p95 is so sensitive to the antiviral state induced by poly-I:C and its replication is blocked it simply can't evolve and what is measured here is the sensitivity to the antiviral state which may be interesting but has nothing to do with the ability to evolve. Show us what happens in the early passages.
In the mouse cell model it is pretty much the same, clearly the design of the experiment is interesting and related to the problematic of evolution but if there is no replication of the BCoV-p95 in the mouse cells it simply cannot have evolution and there is no evidence that the BCoV-p95 replicates in these mouse cells while the wt. virus clearly replicates in those cells and has a chance to adapt to this new environment.
Authors’ response:
For the adaptation experiment of BCoV-p95 in poly IC-induced innate immunity in Figure 5, we thank the reviewer for pointing out the inadequate description. For the first time of infection, 1 MOI was used for both BCoV-p95 and wt BCoV. The titer was 107.7.pfu/mL for wt BCoV; however, almost no virus plaque can be detected for BCoV-p95 (101.4 pfu/mL) at VP0. In order to determine whether it can gradually adapt to the challenge, we performed a blind passage until VP10 and found it cannot replicate since VP1 (please see virus titer results in Figure S4A and RT-qPCR results in Figure S5 in the revised manuscript). Since there is no replication, there is no adaptation. Consequently, we agree with the reviewer’s comments and modified the conclusion as follows: “the replication of BCoV-p95 was blocked by the treatment of poly IC-induced innate immunity” and “BCoV-p95 had no adaptation capability in face of the challenges of poly IC-induced innate immunity”. We also showed which stage of the replication was blocked by poly IC-induced innate immunity (Figure S4 and S5). Together, these have been addressed on page 12, lines 415-422 and Figures S4 and S5.
For the adaptation experiment of BCoV-p95 in alternative host cells (ML cells), we also thank the reviewer for pointing out the unclear description. We in the original manuscript on page 12, lines 416-417, have described that BCoV-95 can replicate although the replication efficiency was relatively low in comparison with that of wt BCoV at VP10. Because BCoV-p95 genomic RNA in ML cells can be detected by RT-PCR, we then can sequence the S protein gene at VP10 as shown in the original and revised manuscript (p9510-ML in figure 5G). Furthermore, because the replication efficiency of BCoV-p95 in ML cells was relatively low in comparison with that of wt BCoV (~0.7%, ~0.8% and ~0.2% vs 100% for genome, S subgenome and N subgenome, respectively) as detected by RT-qPCR in ML cells, consequently, this very low number cannot be shown clearly in the Figure E within a percentage (%) scale. This may cause misunderstanding that BCoV-95 cannot replicate in ML cells. Therefore, based on the results of sequencing analysis from detected genomic RNA by RT-PCR (Figure 5G), and virus titer at VP0-VP10 (Figure S6), it is concluded that BCoV-p95 can replicate and then can adapt to ML cells, but the adaptation capability is lower than that of wt BCoV. To clarify the description and to show it can replicate, we have added the relative value (~0.7%, ~0.8% and ~0.2% vs 100%) in the description and included the detected virus titers in Figure S6. These have been addressed on page 12, lines 440-450; page 13, lines 451-457 and Figure S6.
Sincerely,
Hung-Yi Wu, D.V.M., M.S., Ph.D.
Professor of Graduate Institute of Veterinary Pathobiology
College of Veterinary Medicine
National Chung Hsing University
Tel:886-4-22840369 ; Fax:886-4-22862073 ;Email:hwu2@dragon.nchu.edu.tw

Round 2
Reviewer 1 Report
The authors have sufficiently addressed my earlier comments.
Reviewer 3 Report
I am satisfied with the authors' answers.
I have just one comment on point 2.
- There is some information I would have liked to have about what happens during these 95 passages. All the cells are infected all the time? what about the growth of these cells vs. uninfected cells? what about a possible cytopathic effect or cell death? what happens to the virus produced in the supernatant? is it likely to re-infect persistently infected cells (or are these cells refractory to superinfection?)? in other words can these cells be superinfected with BCoV wt during the passages?
Authors’ response:
During the persistence, cytoplasmic vacuolization and floating cells can be observed. Cytopathic effect can be found occasionally but is not much. Although the shape of the cells is similar between the uninfected and persistently infected cells, the persistently infected cells grow slower. In addition, we also performed superinfection and the results suggest that the persistently infected cells cannot be infected with wt BCoV based on the results of (i) sequencing and (ii) gene expression in which the RNA and protein were not increased with the time of superinfection. We did not test whether all the cells are infected all the time; however, based on the superinfection results, it is speculated that the cells may be almost infected.
Thank you for this information, I think this information is important, and should appear somewhere in the ms because it implies that the viruses produced by these cells do not participate in the evolution of this virus, only a priori the evolution of the intracellular viral population is monitored.